# Identification of Resistance Determinants for a Promising Antileishmanial Oxaborole Series

**DOI:** 10.3390/microorganisms9071408

**Published:** 2021-06-29

**Authors:** Magali Van den Kerkhof, Philippe Leprohon, Dorien Mabille, Sarah Hendrickx, Lindsay B. Tulloch, Richard J. Wall, Susan Wyllie, Eric Chatelain, Charles E. Mowbray, Stéphanie Braillard, Marc Ouellette, Louis Maes, Guy Caljon

**Affiliations:** 1Laboratory of Microbiology, Parasitology and Hygiene (LMPH), University of Antwerp, 2610 Wilrijk, Belgium; Magali.Vandenkerkhof@uantwerpen.be (M.V.d.K.); Dorien.Mabille@uantwerpen.be (D.M.); sarah.hendrickx@uantwerpen.be (S.H.); louis.maes@uantwerpen.be (L.M.); 2Centre de Recherche en Infectiologie du Centre de Recherche du Centre Hospitalier Universitaire de Québec, Université Laval, Québec City, QC G1V 0A6, Canada; philippe.leprohon@crchudequebec.ulaval.ca (P.L.); marc.ouellette@crchudequebec.ulaval.ca (M.O.); 3The Wellcome Trust Centre for Anti-Infectives Research, School of Life Sciences, University of Dundee, Dundee DD1 5EH, UK; L.Tulloch@dundee.ac.uk (L.B.T.); r.z.wall@dundee.ac.uk (R.J.W.); s.wyllie@dundee.ac.uk (S.W.); 4Drugs for Neglected Diseases initiative (DNDi), 1202 Geneva, Switzerland; echatelain@dndi.org (E.C.); cmowbray@dndi.org (C.E.M.); sbraillard@dndi.org (S.B.)

**Keywords:** *Leishmania*, ABC transporters, oxaboroles, resistance

## Abstract

Current treatment options for visceral leishmaniasis have several drawbacks, and clinicians are confronted with an increasing number of treatment failures. To overcome this, the Drugs for Neglected Diseases *initiative* (DND*i*) has invested in the development of novel antileishmanial leads, including a very promising class of oxaboroles. The mode of action/resistance of this series to *Leishmania* is still unknown and may be important for its further development and implementation. Repeated in vivo drug exposure and an in vitro selection procedure on both extracellular promastigote and intracellular amastigote stages were both unable to select for resistance. The use of specific inhibitors for ABC-transporters could not demonstrate the putative involvement of efflux pumps. Selection experiments and inhibitor studies, therefore, suggest that resistance to oxaboroles may not emerge readily in the field. The selection of a genome-wide cosmid library coupled to next-generation sequencing (Cos-seq) was used to identify resistance determinants and putative targets. This resulted in the identification of a highly enriched cosmid, harboring genes of chromosome 2 that confer a subtly increased resistance to the oxaboroles tested. Moderately enriched cosmids encompassing a region of chromosome 34 contained the *cleavage and polyadenylation specificity factor* (*cpsf*) gene, encoding the molecular target of several related benzoxaboroles in other organisms.

## 1. Introduction

Visceral leishmaniasis (VL) is a disease caused by an infection with the protozoan parasites *Leishmania infantum* or *L. donovani* and results, among other symptoms, in serious anemia, wasting and hepatosplenomegaly, which can eventually be fatal when left untreated [1]. The *Leishmania* parasite is transmitted to humans by sandflies, following establishment and differentiation into infective stages in the insect gut [2,3]. In the vertebrate host, parasites infect and replicate inside mononuclear phagocytic cells [4], from where they further disperse to the main target organs: the liver, spleen and bone marrow [5]. The disease is treated with a number of drugs that have disadvantages, such as the need for hospitalization, high cost, toxicity, and the emergence of treatment failure [6]. It is for that reason that the Drugs for Neglected Diseases *initiative* (DND*i*) has invested in the development of novel antileishmanial drugs, with the recent discovery of promising preclinical leads and clinical candidates [7,8]. One of the most promising lead series is the oxaboroles, with DNDI-6148 as the frontrunner, currently in Phase I clinical trials [9,10].

Benzoxaboroles belong to a versatile class of drugs used to treat a wide variety of diseases with low intrinsic toxicity [11]. A viral protease [12], beta-lactamase and leucyl tRNA synthetase (LeuRS) are examples of targets for oxaboroles that have been identified in viruses, bacteria and fungi [13,14,15,16,17,18]. The inhibition of fungal LeuRS relies on the ability of benzoxaboroles to form adducts with the *cis*-diol moieties of sugars, resulting in a stable tRNA^Leu^-benzoxaborole adduct [19]. AN2690, also known as tavaborole, is an FDA-approved example of a benzoxaborole with broad-spectrum antifungal activity [11,20]. The mechanism of action (MoA) has already been evaluated against a number of pathogens [12,13,14,15,16,21,22], including *Trypanosoma* spp. [23,24,25,26,27]. In *T. brucei*, the cleavage and polyadenylation specificity factor 3 (CPSF3) was shown to be a benzoxaborole target [21,22,23,25]. Additionally, one study revealed that the benzoxaboroles needed peptidase-activation for trypanocidal activity [27]. However, the MoA of DNDI-6148 and related oxaboroles in *Leishmania* have yet to be determined.

Drug target identification in kinetoplastids can be achieved through multiple unbiased techniques [28]. In the present study, DNDI-6148 was screened against two *Leishmania* genome-wide, cosmid-based overexpression libraries. The principle behind this approach is that overexpression of a drug target can result in resistance to the corresponding drug by increasing the pool of functional protein. Cosmids containing fragments of *Leishmania* genomic DNA that confer an advantage during compound selection are harvested and then analyzed by next-generation sequencing [29,30]. This analysis allows the overexpressed fragments driving the resistant phenotype to be identified. A frequently used alternative technique is the experimental selection of drug resistance, followed by comparative whole-genome sequencing [28,31,32]. This strategy also provides some information on the propensity toward resistance in clinical settings [33,34,35], which is an emerging issue for almost all currently available antileishmanial drugs [33,34,36,37,38,39]. A resistant phenotype can be acquired in multiple ways, but a common mechanism is an overexpression of ATP binding cassette (ABC) efflux transporters, including ABCB (MDR) and ABCC (MRP) pumps [40,41]. In *Leishmania,* these pumps are implicated in reduced susceptibility to miltefosine (MIL), antimony (Sb) and amphotericin B (AMB) [39,42,43,44,45,46,47,48,49]. Evaluating whether novel compounds are substrates of these efflux pumps, therefore, provides some indication about the emergence of resistance.

Resistance selection and the inhibition of efflux ABC transporters were used in this study as experimental tools to evaluate the likelihood of resistance emergence. Cosmid sequencing supported the identification of resistance determinants, revealing the contribution of genes on chromosome 2 to elevated resistance.

## 2. Materials and Methods

### 2.1. Parasite Cultures

Two laboratory strains, MHOM/MA/67/ITMAP263 (*L. infantum*) and MHOM/ET/67/L82 *(L. donovani*), were routinely cultured at 25 °C in HOMEM (Gibco, Life Technologies) supplemented with 10% inactivated bovine serum (iFBS) (Gibco, Life Technologies, Ghent, Belgium). Ex vivo amastigotes of both strains were purified from the spleens of heavily infected donor hamsters [34]. MHOM/FR/09/LEM4038 and MHOM/FR/96/LEM3323Cl_4_ are two clinical isolates from HIV-infected patients and were only available as promastigotes [50].

### 2.2. Animals

Female Swiss mice and female golden hamsters were purchased from Janvier (Le Genest-Saint-Isle, France) and kept in quarantine for at least 5 days before use. Food for laboratory rodents (Carfil, Arendonk, Belgium) and drinking water were available ad libitum. At the start of the in vivo experiments, hamsters were randomly allocated to experimental units of 3 animals.

### 2.3. Test Substances and Formulations

DNDI-6148 and DNDI-5421 (structures as published in [9]) were provided by DND*i* (Geneva, Switzerland), and stock solutions for in vitro assays (20 mM) were prepared in 100% DMSO. Potassium antimonyl tartrate (Sb^III^) was purchased from Sigma-Aldrich (Diegem, Belgium) and stock solutions were made in phosphate-buffered saline (PBS) at 5.12 mg/mL. The efflux pump inhibitors verapamil, cyclosporine and probenecid were purchased from Sigma-Aldrich and were formulated in 100% DMSO at 20 mM, except for probenecid, which was diluted up to 50 mM in PBS after the addition of ethanol (2%) and NaOH. Dilution series from the stock solutions were prepared in demineralized water to ascertain a < 1% final in-test concentration of DMSO. For the in vivo experiments, DNDI-6148 was prepared at 12.5 mg/mL in 2% ethanol, followed by the addition of 1N NaOH (1.0 eq.) and then further diluted in 5% dextrose in water.

### 2.4. Intracellular Amastigote Susceptibility Assay

Forty-eight hours prior to peritoneal macrophage collection, mice were stimulated by an intraperitoneal (IP) injection of 1 mL 0.2% starch suspension in PBS. After euthanasia with a CO_2_ overdose, macrophages were collected by intraperitoneal lavage with 10 mL of RPMI-1640 (Life Technologies). After counting in a KOVA^®^ chamber, 3 × 10^4^ cells/well were seeded into a 96-well plate in 100 µL of RPMI-1640 macrophage medium, supplemented with 5% iFBS, 2% penicillin/streptomycin and 1% L-glutamine. Macrophages were infected 24 h later with metacyclic promastigotes at a 15:1 multiplicity of infection and incubated at 37 °C and 5% CO_2_ for another 24 h. Next, the culture medium was removed by flicking the plate to eliminate remaining extracellular promastigotes and fresh medium and the drug dilutions were added to the wells. Drug exposure was for a 96 h period without renewal of the medium. Plates were fixated with methanol and stained with Giemsa for microscopic evaluation and EC_50_ determination.

### 2.5. Promastigote Susceptibility Assay

Log-phase promastigotes (±72 h cultures) were counted and diluted to a concentration of 10^5^ promastigotes/well in a 96-well plate to which twofold dilutions of the test compounds were added. Drug exposure was for a 72 h period without renewal of the culture medium, after which, parasite proliferation was assessed using resazurin. Drug activity was determined based on the percentage reduction in parasite proliferation compared to the non-treated control wells.

### 2.6. In Vivo Resistance Selection

Resistance selection was performed as previously described by Hendrickx et al. [34]. Spleen-derived amastigotes of the *L. infantum* strain ITMAP263 were diluted in PBS to prepare an infection inoculum containing 2 × 10^7^ amastigotes in a total volume of 100 μL. Oral treatment through gavage for 5 subsequent days started 21 days post-infection (dpi). Upon relapse, amastigotes were collected from the spleen and used for the infection of new hamsters. These treatment/relapse cycles were repeated for a maximum of 5 cycles. DNDI-6148 was given at 25 mg/kg/day BID, which was shown to result in a > 98% amastigote reduction in both liver and spleen [9].

### 2.7. In Vitro Intracellular Amastigote Resistance Selection

Intracellular amastigote resistance selection was performed as previously described in Hendrickx et al. [33]. In brief, primary peritoneal mouse macrophages were infected with either LEM4038 or ITMAP263 promastigotes in two duplicate 96-well plates and exposed to increasing DNDI-6148 concentrations. One plate was used for Giemsa staining, while the other plate was used for promastigote back-transformation (PBT). The latter entails the release of residual viable amastigotes and allowing their transformation into promastigotes in HOMEM medium at 25 °C. Promastigotes were recovered from the highest drug exposure and further expanded in routine culture. This procedure of infection and PBT cycles was repeated until susceptibility decreased substantially, or for a maximum of five successive passages.

### 2.8. In Vitro Extracellular Promastigote Resistance Selection

Log-phase promastigotes of LEM3323Cl_4_ were counted in a KOVA^®^ counting chamber and were diluted to a final concentration of 5 × 10^6^ promastigotes in 5 mL HOMEM containing the EC_50_ of the oxaboroles. The promastigotes were then left to recover from drug exposure without renewal of the medium. Upon complete recovery, the promastigotes were sub-cultured in a medium containing twice the drug concentration to initiate a next-selection round. This procedure was repeated until the promastigotes could not withstand a higher drug concentration. Finally, an extracellular susceptibility assay was performed to assess the acquisition of resistance.

### 2.9. Intracellular Efflux Susceptibility Assay

Spleen-derived *L. infantum* ITMAP263 or *L. donovani* L82 amastigotes were used to infect peritoneal mouse macrophages at a 5:1 infection ratio. The medium was replaced 2 h later with fresh RPMI-1640 medium containing oxaborole and one of the ABC-transporter inhibitors, namely, verapamil (MDR and MRP inhibitor), cyclosporine A (broad specificity efflux inhibitor), or probenecid (MRP inhibitor). The ABC-transporter inhibitor was added at a single concentration below its EC_50_ which was previously determined [51]; verapamil was added at 8 µM, probenecid at 700 µM, and cyclosporine A at 1.5 µM for *L. infantum* and 2 µM for *L. donovani*. A 4-fold dilution series was prepared for the oxaboroles with the highest in-test concentration of 10 µM. Sb^III^ was used as a reference compound for validation of the assay. EC_50_-values with and without ABC-transporter inhibitor were determined microscopically as described above for the intracellular amastigote assay and the different conditions were compared.

### 2.10. Extracellular Efflux Susceptibility Assay

*L. infantum* and *L. donovani* log-phase promastigotes were diluted to a concentration of 10^6^ promastigotes/well in a 96-well plate, whereafter dilutions of Sb^III^ and oxaborole were added either with or without the ABC-transporter inhibitor. Pump inhibitor concentrations were previously determined and were identical to those in the intracellular assay [51]. EC_50_-values with and without ABC-transporter inhibitor were determined as described above in the extracellular susceptibility assay and the different conditions were compared.

### 2.11. Cosmid Library Primary Screen (L. infantum)

*L. infantum* ITMAP263 promastigotes containing an empty cL-HYG vector were used for initial susceptibility determination [52]. Cos-Seq was performed as previously described by Gazanion et al. [29]. Briefly, *L. infantum* ITMAP263 promastigotes harboring a cL-HYG genomic DNA library were exposed to the EC_50_ of DNDI-6148 in two biological repeats. Parasite growth was followed up daily by spectrophotometry (λ = 600 nm) until parasites reached the late log phase. At that time, part of the culture was diluted and exposed to a higher drug concentration. The remaining culture was used for cosmid extraction and further analysis. This procedure was repeated for several rounds of increased drug exposure.

Cosmids were extracted from the parasites by SDS/alkali lysis followed by phenol/chloroform extraction [53]. RNAse treatment was performed by a combination of 10 µg/mL ribonuclease A and 25 units/mL ribonuclease T1. A second phenol/chloroform extraction allowed genomic DNA digestion with a Plasmid-Safe™ ATP-dependent DNase. Finally, the kinetoplastid DNA was removed by separation on 1% low melting-point agarose gel, followed by a gel extraction and purification of high-molecular-weight cosmid DNA (>50 kb).

Paired-end libraries of each sample were prepared using 40 ng of extracted cosmid DNA and the Nextera DNA sample preparation kit (Illumina, San Diego, CA, USA), following the manufacturer’s recommendations. The libraries were diluted to a final concentration of 8 pM and sequenced with the Illumina HiSeq 2500 system. Gene abundance within samples was quantified using the Kallisto software. Clusters of genes that were significantly enriched by drug selection were retrieved with edgeR using the default parameters (false discovery rate ≤ 0.001). Gene clusters were then plotted according to the median-centered log2 fragment per kilobase per million mapped reads (FPKM) values, using R scripts included in the Trinity package. A first selection was made by only including genes enriched with a log2-fold change of ≥4.

### 2.12. Cosmid Library Secondary Screen (L. donovani)

The second genome-wide cosmid-based library in *L. donovani* has been described previously [30]. Briefly, the *L. donovani* library was maintained at a minimum concentration of 3.33 × 10^5^ cells/mL (1.5 × 10^7^ cells in total) in the presence of 125 μg/mL Geneticin (G418). DNDI-6148 was initially added to the library at a concentration equivalent to 2× the established EC_50_ value (500 nM) before being increased to 750 nM on day 4 of selection. In total, the library was selected with DNDI-6148 for 11 days. At that point, resistant cells were harvested, and cosmid DNA recovered and sequenced using an Illumina HiSeq platform (Beijing Genomics Institute, Beijing, China). Sequence reads were aligned to the *L. donovani* BPKLV9 genome sequence (v46.0, tritrypdb.org). Barcodes flanking the genomic DNA inserts in enriched cosmids were identified with the following sequences: 5′-GCGGCCGCTCTAGAACTAGT-3′ and 5′-CTCTTAAAAGCATCATGTCT-3′ (for fragments in the sense direction) or 5′-ACTAGTTCTAGAGCGGCCGC-3′ and 5′-AGACATGATGCTTTTAAGAG-3′ (for fragments in the anti-sense direction). All associated datasets have been deposited with the European Nucleotide Archive under the following accession number: PRJEB40932.

### 2.13. Cosmid Transfection and Impact on Drug Susceptibility

The cosmids identified from the primary screen were isolated by transfecting the initially extracted pool of cosmids (which was used for sequencing) in high-efficiency competent *E. coli* cells (NEB10). Random colony-picking then allowed the isolation of highly enriched cosmids that were then sequenced for identification. Next, the isolated cosmids of interest were transfected in *L. infantum* LEM3323Cl_4_. 2 × 10^8^ cells were transfected with 10 µg cosmid DNA in a 0.4 cm gap cuvette (BioRad, Hercules, CA, USA) in cytomix buffer. The cells were subjected to two consecutive pulses (10 s) at 1.5 kV, ∞ Ω resistance and 25 µF capacitance, before they were transferred to HOMEM medium. Once parasites recovered (24 h), they were exposed to 150 µg/mL hygromycin. Successfully transfected strains were used for susceptibility testing as described above and for the generation of in vitro growth curves under drug pressure.

RNA was extracted with the QIAamp RNA blood mini kit (Qiagen, Hilden, Germany) from the wild-type strain and the transfected clones. An RT-qPCR was run in duplicate using the one-step SensiFAST SYBR^®^ Hi-ROX kit (Bioline) with primers specific for two genes within the cosmid; LINF_020008700 encoding for a putative proteasome regulatory non-ATPase subunit 6 (primers: 5′-ACGTGAGCAACCTTCTGAGG-3′ and 5′-GCAGCTTGCGATCGAGAATG-3′) and LINF_020008600 encoding for a putative casein kinase alpha chain (primers: 5′-GGATGCCTGTGTGTCCTCAA-3′ and 5′-CAGCGAGCGTAGAATCTCGT-3′) (PCR settings: 1 cycle for 10′ at 45 °C and 2′ at 95 °C, 40 cycles for 5″ at 95 °C and 20″ at 60 °C, followed by a melt curve analysis with an increment of 0.3 °C). The SL gene (Primers: 5′-AACTAACGCTATATAAGTAT-3′ and 5′-CAATAAAGTACAGAAACTG-3′) was used as a reference gene for normalization (PCR settings: 1 cycle for 10′ at 45 °C and 2′ at 95 °C, 40 cycles for 15″ at 94 °C, 15″ at 56 °C, 15″ at 60 °C, followed by a melt curve analysis with an increment of 0.3 °C). The ΔΔCq method was used to calculate the relative normalized expression levels and to evaluate differential expression between the different samples.

Growth curves of the wild-type and transfected promastigotes (harboring an empty or insert-carrying cosmid) were compared when exposed to 2 × EC_50_ and 5 × EC_50_ of DNDI-6148. Promastigote clusters were first separated into single parasites by needle passage (21G × 1½″, 0.8 × 40 mm, 25G × 5/8″, 0.5 × 16 mm) and were then diluted in PBS for flow cytometric (FCM) analysis using the MACSQuant flow cytometer (Miltenyi Biotec). All FCM samples were analyzed in duplicate and further analyzed using the Flow Jo X software. A growth curve was generated by following up the number of promastigotes every 24 h for 7 days, starting with exactly 5 mL HOMEM with 5 × 10^5^ promastigotes/mL.

### 2.14. Statistical Analysis

Susceptibility data and growth curves were analyzed using two-way analysis of variance (ANOVA) and results were considered statistically significant if *p* was <0.05.

## 3. Results

### 3.1. In Vivo and In Vitro Intracellular Amastigote Resistance Selection

In vitro resistance selection for DNDI-6148 was found to be unsuccessful with intracellular parasites, as susceptibility did not change significantly for either the *L. infantum* LEM4038 field strain or the ITMAP263 lab strain (Table 1). For the latter, a small increase in resistance index (RI) could be noted, up to a maximum of 2.5 after the first cycle; however, this was unstable in the following cycles. Similarly, the in vivo resistance selection on ITMAP263, which took approximately two years, did not result in a significantly decreased susceptibility between the wild-type lines (2.95 ± 1.16 µM; 2.56 ± 1.03 µM; 2.57 ± 1.03 µM) and 3 lines of ex vivo amastigotes collected after five successive passages (6.06 ± 2.90 µM; 4.93 ± 1.04 µM; 3.04 ± 0.07 µM).

### 3.2. DNDI-6148 Resistance Selection in Promastigotes

Resistance selection in the promastigote stage against the two selected oxaboroles was found to be difficult and was halted after approximately 60 days. Drug exposure could be increased in a stepwise fashion to a maximum of 2 µM for DNDI-6148 and 48 µM for DNDI-5421 (Figure 1A). Higher drug concentrations led to irreversible growth inhibition and ultimately parasite death. The stepwise selection did not result in a consistent increase of the EC_50_, even though a slight increase could be observed at some passages with an RI < 2 (Figure 1B,C).

### 3.3. The Involvement of Efflux Pumps

The reference compound Sb^III^ was used to validate the assay, as its activity is known to be affected by the ABC-transporter inhibitors verapamil and probenecid. No effects were observed from co-incubating the oxaboroles with the efflux pump inhibitors in either intracellular amastigote or promastigote susceptibility assays (Appendix A).

### 3.4. Screening of DNDI-6148 against a L. infantum Cosmid-Based Overexpression Library

An EC_50_ of 1.44 ± 0.30 µM was determined for DNDI-6148 on promastigotes of the *L. infantum* ITMAP263 line transfected with a whole-genome cosmid library derived from that strain. The transfected parasites could sustain exposure to DNDI-6148 at 1×, 1.5× and up to 2 × IC_50_ (2.88 µM) (Figure 2A). It was not possible to further increase the drug concentration, as parasites were unable to multiply at 4 × IC_50_.

After each selection step, cosmids were extracted from the selected strain and sequenced. Analysis revealed that four cosmids were enriched more than 100-fold (Figure 2B, Appendix A), with one of these cosmids maintaining a fragment of genomic DNA from chromosome 2 (genomic position: 153,221–186,003) being the most enriched (>20,000-fold) (Table 2). Among several other hits identified, a cosmid bearing a fragment of chromosome 34 was enriched by ~770-fold. While less enriched than the cosmid derived from chromosome 2, this cosmid encodes for the *cleavage and polyadenylation specificity factor* (CPSF). Several previous studies have identified CPSF as the molecular target of several related benzoxaboroles in a variety of organisms. The role of this gene and DNDI-6148 MoA in *Leishmania* is being pursued by Wyllie and other collaborators.

### 3.5. Screening of DNDI-6148 against a L. Donovani Cosmid-Based Overexpression Library

In order to rationalize the hits identified in the *L. infantum* cosmid library screen, DNDI-6148 was screened against an equivalent *L. donovani* library [30]. The *L. donovani* cosmid-based overexpression library was screened with DNDI-6148 at 750 nM for a total of 11 days, until the selected population had a similar growth rate to the untreated control (Figure 3A). Cosmids maintained by the “resistant” population were harvested and sequenced, and enriched fragments were mapped to the *L. donovani* BPKLV9 genome. This analysis revealed several fragments of the *L. donovani* genome that were enriched, albeit with relatively low RPKM values (Figure 3B, Appendix A). Interestingly, two genomic fragments enriched in this library screen overlapped with “hits” identified in the *L. infantum* Cos-Seq screening (Appendix A). Specifically, a 62 kb region of chromosome 2 (genomic position 415,500–477,600; Figure 3C) and a 61.5 kb region of chromosome 35 (genomic position 30,326,800–30,388,300; Figure 3D) were identified that encompassed genomic fragments enriched in the primary screen (Table 2). Using the barcodes from this second cosmid library screen, we were able to rationalize these hits to five possible candidate genes on chromosome 2 (Figure 3C, Table 2).

### 3.6. Validation

Since an overlapping fragment of chromosome 2 was enriched in both cosmid library screens, we looked to validate the direct role of this genomic fragment, and the genes it encodes, in resistance to DNDI-6148. Both DNDI-6148 and DNDI-5421 exert nanomolar activity against the intracellular amastigote stage, confirming their potential use as therapeutics. A susceptible strain of *L. infantum* (LEM3323Cl_4_) was transfected with either the empty Cos-Seq cosmid or with the cosmid enriched 20,000-fold in the library screen. Overexpression of the cosmid was confirmed using qPCR, where a 21.6 ± 1.6 fold and a 21.8 ± 1.1 fold increase of expression was recorded for respectively the *casein kinase*
*alpha chain* and *proteasome regulatory non-ATPase subunit 6* genes in LEM3323Cl_4_/CH2 compared to the wild type. The expression of both genes in the strain harboring the empty cosmid (LEM3323Cl_4_/CL) remained unchanged, with respectively 1.4 ± 0.1 and 1.5 ± 0.1 fold expression.

Promastigotes and amastigotes bearing the enriched cosmid (LEM3323Cl_4_/CH2) demonstrated a modest decrease in susceptibility to the two oxaboroles tested compared to those transfected with empty vector (Figure 4). Under drug pressure (2 × EC_50_), promastigotes harboring LEM3323Cl_4_/CH2 also demonstrated a clear growth advantage compared to those harboring the empty cosmid (LEM3323Cl_4_/CL) (Figure 4E). Collectively, these data confirm that elevated levels of the genes encoded on this fragment of chromosome 2, and most likely their protein products, confer an advantage to parasites treated with DNDI-6148 and DNDI-5421.

## 4. Discussion

Screening a library of oxaborole compounds, followed by medicinal chemistry optimization, led to the development of DNDI-6148, a promising antileishmanial compound currently progressing through the various stages of preclinical drug development [9,10]. The MoA of several related oxaboroles has been studied in several pathogenic organisms, such as *T. brucei*, using untargeted metabolomics [20,54], chemoproteomics [21], whole-genome knockdown (RNAi), and overexpression libraries [23,25], and via the selection of drug-resistant parasites, followed by whole-genome sequencing (WGS) [20,21]. To date, no equivalent studies have been carried out for oxaboroles demonstrating antileishmanial activity. Laboratory-selected resistance in *Leishmania* is often obtained by exposing promastigotes and/or amastigotes stepwise to increasing concentrations of drug pressure [28,34]. Although successful for other compounds, in vitro and in vivo resistance selection procedures were unable to generate *Leishmania* clones resistant to our oxaborole compounds. Additionally, the role of efflux pumps as a potential mechanism of resistance to these compounds was evaluated using inhibitors of ABC-transporters such as verapamil, cyclosporine A and probenecid [51,55,56,57,58]. These studies suggest that antileishmanial oxaboroles are not substrates of the evaluated transporters, unlike several other antileishmanials shown to be prone to efflux through these pumps [39,42,43,44,45,46,47,48,49]. Encouragingly, these observations indicate that the oxaboroles studied here may not be prone to rapid resistance development in the field.

In an attempt to identify the molecular target(s) of DNDI-6148 and/or resistance determinants, this compound was used to screen two separate genome-wide, cosmid-based *Leishmania* overexpression libraries. Both screens demonstrated that parasites bearing cosmids that harbored genes from chromosome 2 gained an advantage during selection with DNDI-6148. Indeed, subsequent studies confirmed that promastigotes and amastigotes transfected with cosmid LEM3323Cl4/CH2 were subtly resistant to DNDI-6148 and DNDI-5421, as well as demonstrating a growth advantage under selection with both compounds.

The question remains as to which of the genes encoded on this specific fragment of chromosome 2 is providing an advantage to parasites under pressure from DNDI-6148 and DNDI-5421. Several of the genes present on the selected cosmids have the potential to play a role in the tolerance of treatment with oxaboroles. Previous studies have demonstrated the importance of several of the enriched genes for *Leishmania* survival and life-cycle progression. The proteasome regulatory non-ATPase subunit is a crucial player in protein recycling, vital for the survival and differentiation of kinetoplastids [59,60,61,62,63]. FtsJ-like methyltransferase is generally important for protein biosynthesis [64,65]. In multiple *Leishmania* spp., casein-kinase has been shown to be involved in stress resistance, growth and infectivity [66,67,68]. Given the established functions of the proteins encoded by these enriched genes, it is most likely that one of these plays an indirect role in increased tolerance to oxaboroles, rather than representing their molecular target. Indeed, the mutation of four different enzymes involved in ubiquitination and sumoylation has been observed in *Plasmodium* parasites resistant to benzoxaborole AN13762 [69]. In this study, the mutation of ubiquitination/sumoylation enzymes was thought to be a general response to changes in the parasite stress response rather than an indication of specific drug targets. It is entirely possible that amplification of the ubiquitin-conjugating enzyme E2 (chromosome 2) may be playing the same role in DNDI-6148-treated *Leishmania*. Both cosmid screens pointed to a region derived from chromosome 35 that is enriched in the presence of DNDi-6148. There is an overlap of seven genes in the different cosmids (Appendix A), and one of these genes may also provide a selective advantage in the presence of the drug.

CPSF, established as the molecular target of benzoxaboroles active against *T. brucei* and several other parasites [25], was also identified as a hit in the Cos-Seq library, indicating that the enzyme may also play a role in the mode of action of DNDI-6148 in *Leishmania*. Drug targets are often the most enriched in Cos-seq screens [29]. As other genes were found to be more enriched, it could be argued that CPSF, a 3′ end processing endonuclease, may not be the primary target of oxaboroles in *Leishmania*. However, since this is a target in other species, it is a likely candidate, and work to test it is ongoing in Dundee. It should be noted that this enzyme functions as part of a complex that controls pre-mRNA cleavage, polyadenylation, and transcription termination. Overexpression of a single component of the complex without concomitant rises in the levels of other complex components may significantly limit the levels of resistance achievable via this mechanism. Of course, another explanation for these observations is that antileishmanial oxaboroles might have multiple targets. Further studies will be required to bring clarity to the issue.

In conclusion, our studies suggest that resistance to the promising antileishmanial DNDI-6148 may not develop rapidly upon clinical use, since prolonged exposure to the compound did not lead to the emergence of compound-resistant parasites. Furthermore, DNDI-6148 and DNDI-5421 were confirmed not to be substrates of ABC-transporters, MDR and MRP pumps, a common mechanism associated with resistance to several antileishmanials. The screening of two separate cosmid-based overexpression libraries identified an enriched cosmid, harboring genes from chromosome 2 that provide some advantage under drug pressure. Formal identification of the *Leishmania* target(s) of DNDI-6148 and additional resistance determinants will be important to the clinical development of this compound, and possibly in selecting an appropriate partner drug for future combination therapy.

## Figures and Tables

**Figure 1 microorganisms-09-01408-f001:**
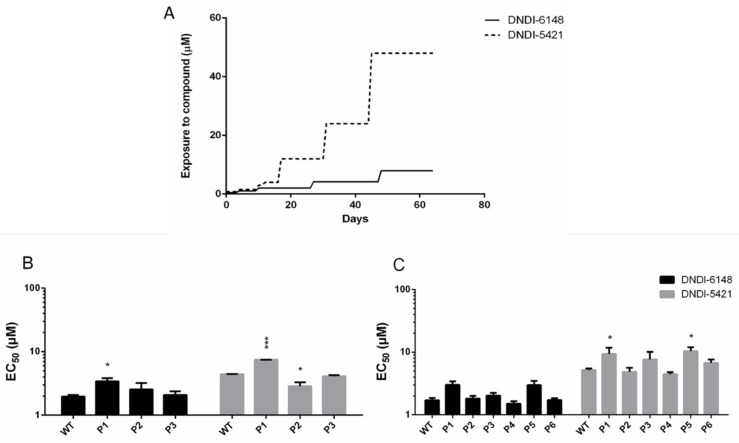
Overview of the stepwise generation of resistant parasite lines toward two oxaboroles (**A**). Comparison of the in vitro susceptibility in the extracellular promastigote assay of the wild-type Scheme 6148 (**B**) and DNDI-5421 (**C**). Results are expressed as the mean EC_50_ (µM) ± SEM, and are based on two biological replicates, each comprising two technical replicates (* *p* < 0.05, *** *p* < 0.001).

**Figure 2 microorganisms-09-01408-f002:**
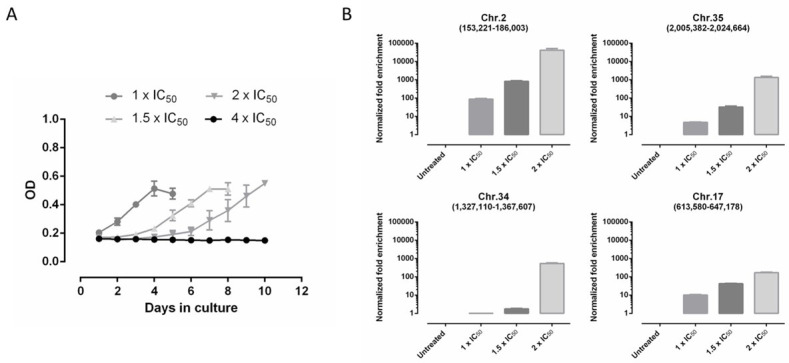
Overview of the Cos-Seq *L. infantum* library selection and bio-analysis. (**A**) Growth curves of parasites transfected with the cosmid library in the presence of increasing concentrations of DNDI-6148. Results are expressed as mean optical density (OD) ± SEM, and are based on two biological repeats. (**B**) Normalized fold enrichment for the most highly enriched cosmids after selection with DNDI-6148. The chromosomal and genomic locations of genomic DNA fragments maintained by enriched cosmids are given.

**Figure 3 microorganisms-09-01408-f003:**
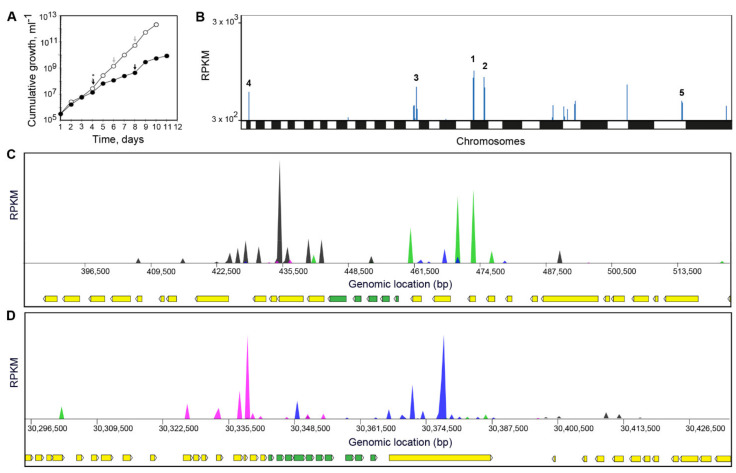
Screen of *L. donovani* cosmid library with DNDI-6148. (**A**) Cumulative growth of the *L. donovani* library in the presence (black circles) or absence (white circle) of DNDI-6148. Arrows indicate where the treated and untreated libraries were passaged. (**B**) Genome-wide map indicating cosmid library hits from the screening of DNDI-6148. The top five hits based on total reads are indicated. (C & D) Focus on two “hits” on chromosome 2 (**C**) and 35 (**D**), respectively. The genes flanked by a majority of barcodes, and therefore most likely to be conferring resistance, are indicated in green, while other genes are indicated in yellow. Possible genes (following a rationalization based on the location of barcodes) are shown as green bars, other genes as yellow bars. The blue/pink and black/green peaks indicate independent cosmid inserts in different orientations.

**Figure 4 microorganisms-09-01408-f004:**
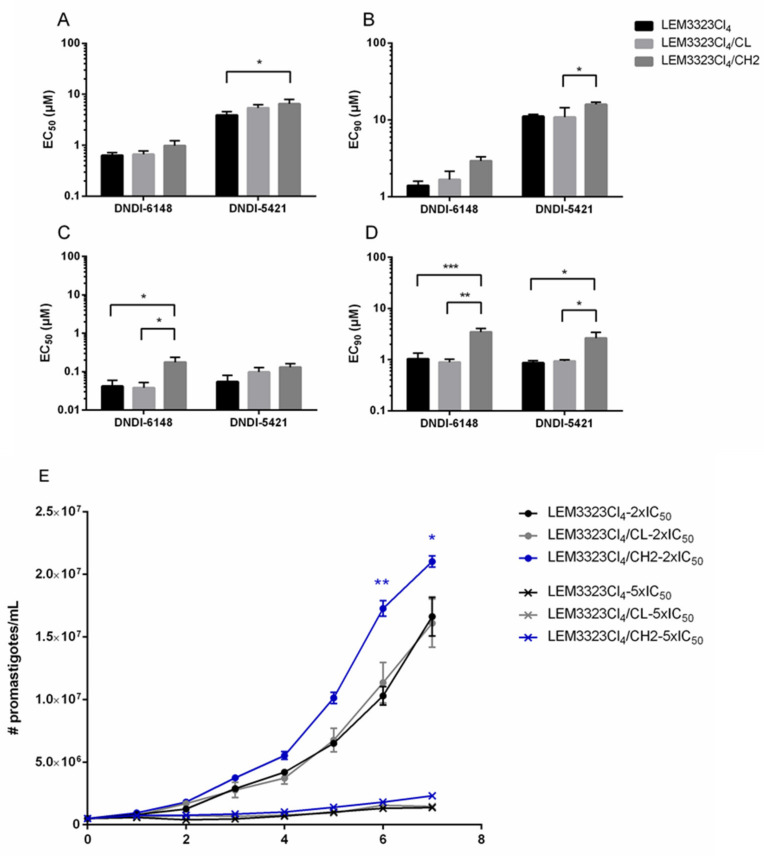
Susceptibility and growth curves of strains harboring either the empty cosmid (LEM3323Cl_4_/CL) or a cosmid harboring genes from chromosome 2 (LEM3323Cl_4_/CH2), compared to the wild type (LEM3323Cl_4_). (**A**,**B**) Promastigote and (**C**,**D**) amastigote susceptibility profiles against DNDI-6148 and DNDI-5421. The EC_50_ (**A**,**C**) and IC_90_ (**B**,**D**) are shown. Results are expressed as the mean EC_50/90_ (µM) ± SEM and are based on at least two independent experiments with biological duplicates. (**E**) In vitro growth curves of the strains exposed to DNDI-6148. Results are expressed as the mean number of parasites/mL ± SEM, and are based on at least two independent experiments run in duplicate. A two-way ANOVA test was performed to test significance (* *p* < 0.05, ** *p* < 0.01, *** *p* < 0.001).

**Table 1 microorganisms-09-01408-t001:** In vitro susceptibility of intracellular amastigotes to DNDI-6148 after each in vitro selection cycle. Results are expressed as the mean EC_50_ (µM) ± standard mean of error (SEM) and are based on two biological replicates, each consisting of two technical replicates.

	ITMAP263	LEM4038
Wild Type	0.24 ± 0.20	0.51 ± 0.11
Cycle 1	0.61 ± 0.20	0.23 ± 0.08
Cycle 2	0.44 ± 0.20	0.21 ± 0.04
Cycle 3	0.48 ± 0.20	0.23 ± 0.06
Cycle 4	0.31 ± 0.04	0.29 ± 0.05
Cycle 5	0.44 ± 0.13	0.27 ± 0.06

**Table 2 microorganisms-09-01408-t002:** Overview of the gene functions of the isolated cosmid-harboring genes from chromosome 2.

Cosmid	Gene ID	Gene Function	Max Enrichment	Rationalised *L. donovani* Gene ID
**Chromo-some 2**	LINF_020008400	EamA-like transporter family/Triose-phosphate Transporter family/UAA transporter family putative	33,225	
	LINF_020008500	Hypothetical protein conserved	25,936	LdBPK.02.2.000320.1
	LINF_020008600	Casein kinase II—alpha chain putative	24,707	LdBPK.02.2.000330.1
	LINF_020008700	Proteasome regulatory non-ATPase subunit 6 putative	30,199	LdBPK.02.2.000340.1
	LINF_020008800	FtsJ-like methyltransferase putative	22,404	LdBPK.02.2.000350.1
	LINF_020008900	Ubiquitin-conjugating enzyme E2 putative	24,949	LdBPK.02.2.000360.1
	LINF_020009100	Mitochondrial protein 81	1852	
	LINF_020009200	Hypothetical protein conserved	44,918	

## Data Availability

Data is contained within the article or Appendix A.

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
