# Peer review of "Identification of Resistance Determinants for a Promising Antileishmanial Oxaborole Series"

_microorganisms, 2021, doi:10.3390/microorganisms9071408_

Round 1

Reviewer 1 Report

This is an interesting manuscript addressing important aspects of drug resistance development using oxaborole compounds.  I have a few comments for the authors to address to make this a stronger manuscript:

  1. in the introduction please commend more about the use of oxaboroles as anti-fungal drugs since this was the first use of this type of compounds and would give the reader a stronger sense of the various oxaboroles used in treating pathogens and some side effects that have been reported. This is of interest since the references identified by the authors (11-22) indicate a wide variety of affected organisms but little on ‘side-effects’ relative to host.
  2. Please provide the structures of the two used oxaboroles used in this study (DNDI-6148 and 5421) so that the reader can get a sense of how previous oxaboroles differ; this will help put the various organisms targeted into some perspective. The authors can also speculate if the mechanism of action of these two compounds is similar.
  3. please define abbreviations first time used; for examples lines 68, 202 (what is G418?), 344 (WGS ?).
  4. where is the ethical (IRB) statements especially since the authors are using clinical isolates
  5. lines 110-111: how was the culture medium renewed? Please provide more information
  6. Table 1 add missing ± to last column bottom
  7. For figure 2A: at what wavelength were the values collected? Also for this figure, line 274 indicates up to 2 x IC50 value but in the graph there is also a line 4 x IC50
  8. The authors should comment (speculate?) on the difference in sensitivity in the responses (Figure 4 A, B, C, D) between the promastigotes and amastigotes and thus any potential role in therapeutics.
  9. The format for the reference section is not consistent (titles are sometimes in caps and sometimes not)

Reviewer 2 Report

The manuscript of Van den Kerkoff et al. describes a genomic screen to identify possible targets for the anti-Leishmania oxoborole drugs. The authors search for resistance to these drugs using several approaches. First, they attempt to generate resistance directly to the drugs through multiple passages, and were unable to do so; this negative result is interesting in that it suggests difficulty in avoiding drug action by Leishmania. They then combine screening a genome wide Leishmania cosmid library with inhibition of drug efflux transporters. Their hypothesis is that over expression of a host target would lead to resistance by increasing the concentration of target molecules. Using this approach they find a cosmid corresponding to a region of chromosome 2 consistently enriched in two different strains, as well as a region corresponding to CPSF (cleavage an polyadenylation specificity factor, which processes 3’ ends of RNA pol II transcripts). The top hit in chromosome 2 contains some interesting targets, including a proteasomal subunit, a kinase and E2 ligase. The experiments are carefully performed and interpreted. However I would have liked to see deeper experimentation, and presentation to merit publication as outlined below.

  1. The structures of the drugs used, and a broader discussion of the mechanism of action of this class of drugs should be presented in the introduction. Are these pro drugs? Do they target cis diols? The authors should see the nice review by Benkovic on this matter.

  1.  The authors should demonstrate that in cases of cosmid enrichment the actual gene targets are enriched, preferably by Western blots.

  1. For the potential targets, expressing enzymatic null mutants would have provided addition insights to these targets: the E2 ligaes, kinase and CPSF all can be readily mutated in their active sites. They should at least attempt this experiment on one target.

  1. If CPSF is a target, this is readily tested in vitro. Has this been done? it’s straightforward. Also, the authors could show altered RNA processing in presence of drug using RNA seq. 

    The manuscript of Van den Kerkoff et al. describes a genomic screen to identify possible targets for the anti-Leishmania oxoborole drugs. The authors search for resistance to these drugs using several approaches. First, they attempt to generate resistance directly to the drugs through multiple passages, and were unable to do so; this negative result is interesting in that it suggests difficulty in avoiding drug action by Leishmania. They then combine screening a genome wide Leishmania cosmid library with inhibition of drug efflux transporters. Their hypothesis is that over expression of a host target would lead to resistance by increasing the concentration of target molecules. Using this approach they find a cosmid corresponding to a region of chromosome 2 consistently enriched in two different strains, as well as a region corresponding to CPSF (cleavage an polyadenylation specificity factor, which processes 3’ ends of RNA pol II transcripts). The top hit in chromosome 2 contains some interesting targets, including a proteasomal subunit, a kinase and E2 ligase. The experiments are carefully performed and interpreted. However I would have liked to seen deeper experimentation, and presentation to merit publication as outlined below.

    1. The structures of the drugs used, and a broader discussion of the mechanism of action of this class of drugs should be presented in the introduction. Are these pro drugs? Do they target cis diols? The authors should see the nice review by Benkovic on this matter.

    1.  The authors should demonstrate that in cases of cosmid enrichment the actual gene targets are enriched, preferably by Western blots.

    1. For the potential targets, expressing enzymatic null mutants would have provided addition insights to these targets: the E2 ligaes, kinase and CPSF all can be readily mutated in their active sites. They should at least attempt this experiment on one target.

    1. If CPSF is a target, this is readily tested in vitro. Has this been done? it’s straightforward. Also, the authors could show altered RNA processing in presence of drug using RNA seq. 

Round 2

Reviewer 2 Report

I think that the paper can be accepted in its present form.